# The Complex Bridge between Aquatic and Terrestrial Life: Skin Changes during Development of Amphibians

**DOI:** 10.3390/jdb11010006

**Published:** 2023-01-30

**Authors:** Esra Akat Çömden, Melodi Yenmiş, Berna Çakır

**Affiliations:** Department of Biology, Faculty of Science, Ege University, 35040 Izmir, Turkey

**Keywords:** metamorphosis, tadpole, amphibian, skin, chromophore, glands

## Abstract

Amphibian skin is a particularly complex organ that is primarily responsible for respiration, osmoregulation, thermoregulation, defense, water absorption, and communication. The skin, as well as many other organs in the amphibian body, has undergone the most extensive rearrangement in the adaptation from water to land. Structural and physiological features of skin in amphibians are presented within this review. We aim to procure extensive and updated information on the evolutionary history of amphibians and their transition from water to land—that is, the changes seen in their skin from the larval stages to adulthood from the points of morphology, physiology, and immunology.

## 1. Introduction

Amphibians have undergone behavioral, morphological, and lifestyle-based evolutionary processes that have tolerated their existence for millions of years, before and after dinosaurs [1]. Extant amphibians, i.e., Lissamphibia, with more than 6400 species consist of three distinct orders, Anura (frogs), Caudata (salamanders), and Gymnophiona (caecilians) [2]. Lissamphibia emerged nearly 282 million years ago in the Early Permian period and the three orders diverged nearly 251 million years ago. The relationships of the three orders were first explained by the Procera hypothesis, which supported a close relationship between caecilians and salamanders. However, recent studies, including fossil research, are in favor of the Batrachia hypothesis, which supports a close relation among frogs and salamanders [3] (Figure 1).

The order Anura, i.e., tail-less amphibians consisting of frogs and toads, covers more than 90% of extant amphibian diversity. Major clades of Anura emerged in three periods: Triassic, Jurassic, and early Paleogene. The large quantity of the anurans has a typical biphasic life history, i.e., their reproduction and development depend on water presence. However, a good deal of lineages inclines towards terrestrial life in either the larval or adult stages. The order Caudata, i.e., salamanders, includes the second highest species diversity. Different from other adult amphibians, they have a body plan that has barely changed since the Jurassic period, with a tail in all phases, limbs, teeth on both maxilla and mandibula, ribs, and a number of skull bones. The order Gymnophiona, i.e., caecilians, have a snake-like body plan with no limbs and reduced tails. Male individuals use the bivious cloaca for copulation. Even though direct development is common, some members still have an aquatic larval phase, which is an ancestral trait. They have a segmented skin covered with dermally embedded scales within pocket-like pits [4,5,6].

Amphibians are the first tetrapods among extant animals that adapted to terrestrial life coming from aquatic environments [7]. They have adapted to a ranging multeity of habitats as animals with pervious, bare skin that must be preserved as mildly moist for cutaneous respiration and to be able to sustain their body temperature just above the environmental temperature [8]. They have adapted greatly in terms of morphological and physiological traits, which have been conserved throughout the taxon such as the skin with multifarious adaptations for new environmental provisions. This transition has been a climacteric period in vertebrate evolution [9]. This situation is a gold mine for researchers because it enables them to study both aquatic and terrestrial habitats.

Frog integument has some distinctive traits such as water loss prevention, water intake by aquaporins, or alkaloid defense, which can be followed back to fish skin [10]. What distinguishes amphibian skin from their piscine progenitors is the thin keratinous stratum corneum and the formation of a corneous cell envelope, a sign for evolution to terrestrial life [11]. There are many dermal glands (mucous and granular) with various functions that tell amphibians apart from other vertebrates [12,13]. It has been suggested that a chemical defense system of the amphibian integument that is regulated by molecules, either arose in the early evolution of Anura or emerged convergently within Anura [14]. As for skin coloration, it is known that eumelanin and pheomelanin are mostly conserved among vertebrates, therefore it can be argued that they are responsible for the brown-black coloration in both extinct and extant amphibians, even though melanin-based color is hard to determine in fossil amphibians and basal archosaurs [15]. Another substantial adaptation through amphibian evolution is the change of the developmental model from the typical indirect way (with metamorphosis) to the direct way (the loss of an aquatic larval phase, i.e., development without metamorphosis), which is more suitable for a lifestyle change. It has been argued that the loss of the aquatic larval phase evolved several times independently in consistency with the patterns of parallel evolution [16,17].

During amphibian metamorphosis, the body changes rapidly and dramatically. Among other organs, the skin of amphibians exhibits new modifications for a new life [18]. Superficial and keratinized epidermal cell layers and multicellular exocrine glands contribute to protection against dehydration and a number of unfavorable effects of the terrestrial environment [7,19]. Information related to skin changes during the development of amphibians is crucial to uncover the mechanisms of how vertebrates survive and evolve. This review aims to present what changes occur in the skin during the development of amphibians and the transition between aquatic and terrestrial life with original skin sections and illustrations.

## 2. Skin Changes during the Transition from Larva to Adult

Metamorphosis is composed of three periods (a) *pre-metamorphosis*, larval growth, (b) *pro-metamorphosis*, preparation for metamorphosis, and (c) *the metamorphic climax*, which is considered as completion metamorphosis [9]. Skin is one of the organs that has experienced the most extensive metamorphic rearrangement. This redesign is characterized by the degradation of the uppermost epidermal cells and the proliferation of forming the adult epidermis [17,20]. When amphibians metamorphose from tadpole to frogs, the transformation of the thin larval skin to adult multi-strata skin is one of the key alterations to adapt to the drier habitats [21,22] (Figure 2).

Ciliated larval epidermal cells begin to disappear after hatching [24]. The mucociliary role of these cells in frogs is similar to other organisms, and the cilia act in a prominent role in removing microbial organisms from mucous-covered surfaces. The larval skin is composed of 1–2 layers of basal cells that are covered by apical cells [25]. Epidermal basal cells are lined beneath the club cells, which are prominent components of the epidermis in larval amphibians. These cells are uniquely differentiated components of larval skin and they have an important physiological role in the production and storage of fluid and mucous materials. Club cells have been referred to as “mucous cells, Leydig cells” [26]. The epidermis of the larvae also includes an outermost layer of a mucus-like secretion including acidic glycoconjugates [27].

The larval epidermal cells progressively disappear and are switched by squamous cells during the metamorphosis. Mucous and serous glands arise from similar cellular nests or buds, which are formed through the division of basal cells in the epidermis [28]. The basal membrane is beneath them, which is followed by the stratum compactum that consists of a dense connective tissue. Some dorsal epidermal cells include melanin pigments obtained from melanophores but not the ventral skin. Collagen fibers of the compact layer are more loosely packed than in the previous stages. This provides the migration of mesenchymal cells in the subcutaneous tissue between the fibers and forms the stratum spongiosum. This new layer is thicker in dorsal skin than ventral skin, particularly in dorsal ridges or dermal plicae. Additionally, in early stages, melanin is present in both the epidermis and subcutaneous tissue of the dorsal body. Melanophores and iridophores are present in the spongious layer of the dorsal, which includes gland nests and some small glands as granular and mucous. However, ventral skin includes only gland nests, pigment cells or glands that are not visible [29]. Granular and mucous glands already showed a neck and a short duct at the end of metamorphosis [27].

Up to stage G30 of *Rhinella arenarum*, skin is composed of a one-layered epidermis that has cells with heterochromatic nuclei and no glandular cells. Dermis, where melanophores are present, is a thin layer. At this stage, two-layered epidermis consists of two different types of glandular cells that are dispersed, which are mucous and unicellular secretory cells. At stage G42, the skin of this species is composed of two or three epidermal layers and a thicker dermis. Stratum spongiosum becomes observable as a thin layer, in between the epidermis and the stratum compactum. Granular and mucous glands are already displayed in a short duct, ultimately at the metamorphic climax [27] (see Table 1 for the summary of all the species discussed within this review).

When it comes to skin development of the *Arthroleptis* genus, the skin is composed of a single-layered epidermis. Large and cuboidal epidermal cells have globular nuclei in the dorsal region of the body, while they are flat with ovoid to spindle-shaped nuclei in the ventral region of the body. The cells have intracellular yolk granules, as well as dark-pigmented vacuoles. Ciliated cells are present in a distributed arrangement throughout the embryo. A small number of melanophores are present among epidermal cells. A few unicellular mucus glands are noticeable [17].

At stage G27 of *Cardioglossa sp.*, the arrangement of epidermis is typical as in most tadpoles. The bilayered epidermis in which the apical cells have different shapes in different body regions (e.g., flat in the head, sizeable in the trunk regions) is followed by the basal layer, which is composed of cuboidal cells that have globular nuclei. Unicellular gland cells lie in the epidermis, from the basal layer to the surface. Stratum compactum, which is related to melanophores, is found underneath the epidermis as a thick layer. The number of melanophores decline dorsoventrally in both the head and trunk [17,30].

Typical larval epidermal cells of *Hynobius retardatus* disappear in the metamorphosis and are substituted by adult epidermal cells. When metamorphosis is arrested by distinct processes such as goitrogen treatment, thyroidectomy, hypophysectomy, and growing at low temperature, club cells remain completely. However, dermal glands behave dissimilarly from club cells due to the fact that dermal glands develop and differentiate from basal epidermal cells. The death of the organs and/or cells that are specific to larvae is followed by the replacement or phenotypic transformation of adult ones [28].

Amphibians undergo major changes through their transformation and adaptation to land, e.g., the skin of larvae degenerates rapidly and remodels a multistratified glandular epidermis containing a hard stratum as a result of keratinization and some cornification [31]. Metamorphosis is a phase with multifarious morphological and biochemical changes that are controlled hormonally, such as thyroid hormones, prolactin, adrenocorticotropic hormone, and corticoids [28]. 

The epidermis of an adult is composed of a stratified squamous epithelium and the dermis of dermal glands. There are four consecutive layers within the epidermis: the germinative layer (st. germinativum), spinous layer (st. spinosum), granulous layer (st. granulosum), and hard layer (st. corneum), from inside out, respectively [32]. The hard layer (st. corneum) is formed by keratinization and is a thin layer. Keratinization is a terminal differentiation process of degradation and loss of the nuclei that results in skin remodeling as the amphibian undergoes metamorphosis [33]. The dead keratinocytes of the uppermost keratinized layer are nucleus-free and filled with keratins. Among keratin protein content of the keratinocytes, one protein is conserved among anamniotes as well as reptiles, Krt117. The absence of this protein among avian and mammal epidermis suggests that the skin of these taxa took form through gene loss. 

Epidermal keratinization of amphibians was vital through terrestrial adaptation among aquatic life because it provided osmo- and thermoregulatory systems and respiration as well as protection against mechanical damage. Amphibians produce peptides that were recently found to be antimicrobial or antioxidant, and this discovery led to clinical research on wound healing, regeneration, and bioactive peptide production throughout the world as a rising study topic [34].

Both keratinocytes and granular cells have tight junctions in between [35]. In contrast to corneocytes in the skin of amniotes, stratum corneum of the amphibians consist of dying keratinocytes, which make this layer water permeable, thus the role of water barrier is left to inner granular cells which are named “first-reacting cells” [36]. The granulous layer consists of cells that are rich in mitochondria, in which AQP-h2 and AQP-h3 were detected via immunofluorescence [37].

The stratum spinosum consists of terminally differentiating cells lying between the hard and germinative layers, which includes distinct cell types such as epithelial cells, immune cells, and chromatophores that carry out pigmentation patterns. The dermis consists of two different layers: spongious and compact dermis. The spongious dermis consists of loose connective tissue [38] (Figure 3). 

The subdermal region of amphibian skin (subcutis or tela subcutanea) is a thin layer of vascularized loose connective tissue, which commonly includes some adipocytes, and the hypodermis forms the outer lining of the subcutaneous lymphatic space (Figure 2). Additionally, hypodermis may include iridophores as in *Hyperolius nasutus* and *Hyperolius viridiflavus* [39,40]. Moreover, some of the water-transporting proteins (aquaporins) are present in the hypodermis of *Dryophytes chrysoscelis* (formerly *Hyla chrysoscelis,* Cope’s gray treefrog) [41] (Table 1).

## 3. Chromatophores and Coloration

The chromatophores are pigment-containing cells that color the skin of reptiles, amphibians, and fish [9]. They originate from the neural crest and provide the varied color tones of different amphibians, making them responsible for camouflage, protection, and signaling, which may result in the secretion of toxic poisons. Chromatophores are divided into three types: xanthophores, iridophores, and melanophores [42].

Xanthophores are found mostly in the chromatophore unit in the dermis. This chromophore type includes pteridine pigments, which provide yellow, orange, or red coloration. Iridophores (guanophores) that lie below xanthophores are reflective cells consisting of platelets that contain purine pigment as iridescent material [9,43]. Although some melanophores have an epidermal localization, most of them are situated in the dermis (Figure 4A,D). 

Although animals have improved color pattern matching for their substrate as a main adaptation, they may be exposed to heterogeneous environments. Therefore, one type of pigmentation model would not support camouflage for all the alterations in the surrounding environment. Many amphibians need cryptic coloration as their main defense. Amphibians, along with structural adaptations, may be camouflaged by phenotypic plasticity in skin coloration. Depending on environmental conditions such as background color, light intensity, humidity, stress or temperature, melanosomes can be rearranged in the melanophores for lightening or darkening of the skin. While these color changes are not as sudden as in true disguise masters such as chameleons, amphibians may change colors relatively rapidly to inhibit detection by predators and these changes may last for weeks [44]. Commonly, the iridophores constitute a single layer (Figure 3), yet some species (e.g., the African frog, *Chiromantis petersi* (Rhacophoridae)) may have three to five layers of iridophores. In *C. petersi,* the dermal chromatophore unit with multilayered iridophores increases the blue light reflection, making the animal appear red [9,45].

Even though human melanophores contain a complexion of two pigment types, eumelanin (black-brown) and pheomelanin (red-brown), amphibian melanophores have only the former and lack the latter. With the dendritic processes and the pigment content within, melanophores are basic cells for black and/or brown coloration [46,47]. 

When the melanosomes are aggregated around the nucleus, the skin appears light-colored. If melanosomes are widely distributed along all cytoplasmic extensions, the skin looks darker. Melanophores and melanosomes are great models for understanding molecular mechanisms of intracellular transport. Along the microtubules, melanosomes are transported by both kinesin and dynein motors. Inhibitory and stimulatory hormone mechanisms on chromatophore units seem quite complex. Melanophores are controlled by multifarious hormonal stimuli such as α-MSH, noradrenaline, vasoactive intestinal polypeptide, calcitonin gene-related peptide-β, melatonin, adrenaline, acetylcholine, histamine, serotonin, and endothelin. The two most researched hormones on this subject are α-MSH and melatonin. According to a study with *Xenopus laevis*, melatonin acts prior to metamorphosis; after metamorphosis, α-MSH handles the melanophore depending on the background light intensity [9,48].

## 4. Amphibian Gland Formation, Types, and Function

Ectoderm-derived progenitor cells of glands in anurans start to aggregate in premetamorphic stages. Then, in the initial metamorphic stages, these cells form an alveolus including early granules and turn into small adult-type glands with secretory components during metamorphic climax. Skin glands move from the epidermis to the dermis during the terrestrial juvenile phase [7]. It has been reported that [49] *Phyllomedusa bicolor* tadpoles had mucous and granular glands; however, the gland duct was not developed until metamorphosis. Granular glands of amphibian skin may not be able to excrete antimicrobial peptides fully on the skin surface until development of the neuromuscular secretory mechanism and gland ducts in the epidermis [50].

Primarily, two alveolar (acinar) gland types are present in amphibian skin, mucous and granular (serous) glands (Figure 4A,C,D). Other types are detected in some anurans. Dermal glands are composed of three parts: the duct, the intercalary region, and the gland alveolus. Keratinocytes that are modified make the duct, which is the way to transport the secretion through the skin (Figure 4C). The gland alveolus in the spongious layer contains secreting epithelium. In between these two structures, there is the transitional intercalary region [7].

In some body regions of several amphibians, a group of reproduction- and defense-related granular glands is found called macroglands. These glands are divided into five types: parotoid, lumbar, hedonic, paracnemid, and pectoral. Parotoid macroglands are present in several urodeles and anuran species (Figure 4B). The paracnemid macroglands are located around the tibia in some anurans. In *Bufo*, they histologically resemble the parotoid macroglands. In some leptodactylids, they are located right behind the sacrum and called lumbar or inguinal macroglands. They probably function in defense. The parotoid and inguinal macroglands seem to have a significant part in defense [7]. Additionally, the position of the parotoid gland has strategic importance to survive. For example, in a slow-moving amphibian such as *Salamandra infraimmaculata*, the localization of the parotoid glands probably prevents predators from biting the head.

The mucous glands are smaller (Figure 4B) but more numerous than granulous glands [51]. The adenomere (the secreting part) of mucous glands, which secrete mucus that will cover the surface of the skin, are made of cubic or cylindrical epithelia. Mucus contains various “frog integumentary mucins” (FIMs), one of which is the threonine- and carbohydrate-rich FIM-A.I. [52]. The mucous glands secrete mucus, which acts in vital functions such as cutaneous respiration, thermoregulation, reproduction, and defense. In one member of the Ranidae (*Lithobates pipiens*), it has been found that this covering mucus contains glycol (9%), sulfate (0.4%), and carboxyl groups alongside the glycosaminoglycans (16% proteins, dry weight). Mucus has vital roles such as pH and moist preservation, as well as thermoregulation, lubrication, and adhesion. Some amphibians also take advantage of the mucus in terms of protection because it includes toxic compounds and makes the body slippery [53].

As fundamental components, the toxic ingredients of the granular glands are defensive against many organisms. The granular glands are mainly composed of a gland alveolus with a secretory cell line and outmost myoepithelial cells. The contractile myoepithelial layer is involved in the secretion of glandular products (Figure 3A). Although products of granular glands are toxic substances, recent studies show their variety of pharmacological effects and other functions. Granular gland secretions generally include biogenic amines, peptides, steroids, alkaloids, and guanidine derivatives. These secretions may have pharmacological influences, such as being anesthetic, hypertensive, hypotensive, myotoxic, neurotoxic, cardiotoxic, and hemotoxic [7].

## 5. Skin Immunology

Amphibians are commonly studied animals because their skin serves as an indicator of environmental pollution. Even though pollution and habitat damage are significant threats to amphibians, the main reason for the decrease in the number of these animals is infection by *Batrachochytrium dendrobatidis*. This type of fungus colonizes and grows in epidermal cells, causing death by chytridiomycosis [54,55].

Amphibian skin is one of the significant natural antimicrobial peptides (AMP) resources. AMPs generated by a variety of vertebrates, invertebrates, and plants are an evolutionarily preserved constituent of the host’s innate immunity [56]. In addition to their main role of protection from harmful microorganisms invading the host, they also have roles involved in host immune modulation, such as endotoxin neutralization, chemotaxis, and healing injuries [57]. 

AMPs have distinct specificities. Some of them kill only bacteria, some of them kill only fungi; however, some of them are active against a broad spectrum of pathogens such as bacteria, yeasts, fungi, protozoa, and viruses. These peptides play a role in the innate amphibian immune response and act as adaptive immunity against pathogen infiltration [4]. AMPs are attracting attention as potential therapeutic agents against pathogens such as bacteria and fungi that show resistance to traditional antibiotics [58].

Bioactive molecules contained in granular (serous) glands and categorized as AMPs have broad-spectrum antimicrobial activity. The main group of amphibian AMPs are brevinins, temporins, dermaseptins, japonicins, magainins, esculentins, nigrocins, palustrins, ranatuerins, ranalexins, cathelicidans, and tigerinins. Not all AMP classes are expressed in the skin of all amphibian species. For instance, *Xenopus laevis* harbors four different AMP families: caerulein precursor fragment (CPF), peptide glycine-leucine-amide (PGLa), xenopsin precursor fragments (XPF), and magainins [38].

The granular glands are completely developed only after metamorphosis, which may show that the innate defenses of larvae are insufficient. These findings force the issue of how tadpoles’ skin is shielded from microbial pathogens before metamorphosis starts. Some experiments have demonstrated that the granular glands of tadpoles have the ability to secrete peptides [59]. It has been shown that [60] production of skin peptides in tadpoles of *Ranoidea splendida* (formerly *Litoria splendida*) begins long before the onset of metamorphosis. However, analyses of skin peptide activities against *Escherichia coli* (ATCC 25922) in all three stages in frogs (larvae, metamorphic stage, and adult) showed that the innate immune system is not developed in the first stage but is present in the last two stages. This result is consistent with the findings of small granular glands in tadpoles and uperin 7.1-rich antimicrobial peptides of metamorphs and adults. As stated above, metamorphic and adult individuals have antibacterial peptides, which might indicate a specific adaptation to a distinctive microbial flora. Having this activity must be vital for metamorphs because metamorphosis-related corticosteroids repress the innate immune system through a transitional stage in which they have to adapt to both water and land [59]. All of these findings lead to common ground about AMPs, which is that they are vital for protection against pathogens with additional roles such as healing skin injuries, endotoxin neutralization, and chemotaxis [57].

Amphibians have colonized many different habitats around the world such as terrestrial, arboreal, and semi-aquatic environments. Among them, semi-aquatic habitats are predisposed to pathogens much more than the first two because water is a fine transmitter for certain pathogens such as *Batrachochytrium dendrobatidis*. Therefore, a good bit of AMPs is secreted in the Ranidae skin (e.g., *Odorrana margaretae, Pelophylax nigromaculatus* etc.) (Table 1). Nonetheless, terrestrial and arboreal amphibians also have protection mechanisms of different kinds that made the adaptation to complex habitats possible [59].

AMPs, which are amphipathic and cationic in nature, procure a secondary α-helical conformation during microbial contingence. For instance, cationic, leucine- and isoleucine-rich Figainin 1 has eighteen amino acid residues and is active against Gram-negative bacteria and epimastigote forms of *Trypanosoma cruzi*. Figainin 1 also exhibits cytolytic effects against human erythrocytes as well as antiproliferative activity against cancer cells. Although it has adverse effects on noncancerous cells, this peptide shows interesting features for the improvement of new drugs against cancer and infections [61].

Not only is secretion produced from granular glands, but mucus also has a significant role in the physical and chemical defense versus pathogens. A study on the epidermis of *Xenopus tropicalis* tadpoles designated the development of multiciliated cells, ionocytes, goblet cells, and small secretory cells as integral to mucosal barrier formation [38]. Moreover, hyaluronic acid (HA) and dermatan sulfate are significant components in the dermis of various amphibian species. HA is an important component of the interstitial barrier. It maintains the reduction of tissue permeability through the higher viscosity of tissue. Enzymes, such as hyaluronidase, are capable of hydrolyzing HA, and the hydrolysis of HA causes increased tissue permeability. For this reason, hyaluronidase is applied in combination with some drugs to accelerate their delivery and dispersion. Several bacteria, such as *Clostridium perfringens*, *Streptococcus pneumoniae*, and *Staphylococcus aureus*, are able to increase mobility through tissues in the body via producing hyaluronidase to hydrolysis of HA [62,63,64,65]. Literature data reveal that HA is a protective barrier against chemicals and pathogens. Therefore, HA was mainly located in the upper region of the dermis beneath the epidermis of *Hyla savignyi* and *Salamandra infraimmaculata* (Figure 5A,B) (Table 1).

## 6. Water Relations

The skin–water relation is life sustaining for amphibians because their endurance depends on the moisture rate of the skin. As a disadvantage to this thread, water evaporates very fast from the surface of amphibian skin, which was thought to be an edging feature of the body thermoregulation. Even though there are ways to conserve body temperature through different body posture or opting for new microhabitats, thermoregulation is never guaranteed. However, some species have skin that is resilient to water-loss and are able to regulate this resilience by physiological processes [66].

Some amphibians display rather different adaptations to dry states. They have dermal glands that produce and pour lipids to the skin surface to reduce evaporative water loss [67]. In most amphibians, water intake is governed by the skin through aquaporin-formed channels instead of drinking water by the mouth. Aquaglyceroporins (AQPs) are expressed in many epithelial cells, including the epidermis, maintaining water transmittance through the cell membranes [68]. In *Hyla japonica,* for example, two types of aquaporins (AQP-h2 and AQP-h3) are induced by arginine vasotocin (AVT), an anuran homologue of mammalian vasopressin. Additionally, in semi-aquatic Ranidae members (e.g., *Rana japonica, R. nigromaculata*, and *R. catesbeiana)*, the ventral skin of the hindlimbs reacted to AVT by increasing AQP-h3 expression. On the other hand, aquatic *Xenopus laevis* that have to minimize water intake stayed indifferent to AVT [9].

As osmoregulatory organs, the ventral skin’s pelvic patch and urinary bladder are vital. AQPs obtained from both amphibians and mammalians were subjected to phylogenetic analyses and the results showed that there are six types of it in the anurans: types 1, 2, 3, and 5, and anuran-specific types a1 and a2, among which the last two were found to be in the skin of the urinary bladder. Additionally, AQP-h2-like protein has been found in the ventral skin’s pelvic patch of terrestrial and arboreal anurans, which might indicate that this specific protein emerged during the adaptation of amphibians through drier habitats [69].

The dermis of terrestrially spread adult amphibians is divided into spongious and compact dermis (Figure 3) through the presence of the Eberth–Katschenko layer. This noncellular layer completely consists of glycosaminoglycans and glycoconjugates in which hyaluronic acid and dermatan sulfate are key components in various species. Hyaluronic acid and other hyaluronic acid-like molecules in the Eberth–Katschenko layer are mainly present in the dorsal side of amphibian skin [38]. HA is a high-molecular-weight glycosaminoglycan that has prominent biological functions from bacteria to high vertebrates including humans. The HA has an ability to bind large volumes of water, and this feature makes it a perfect lubricator and water absorber. It has been reported [64] that the HA is one of the most hydrophilic molecules in nature, and it has been defined as nature’s moisturizer. Water-retentive hyaluronic acid helps to decrease water vaporization, thus assisting to prevent desiccation, especially in basking amphibians [70,71]. HA immunoreactivity is mainly observed in the dermis, and it was mainly distributed in the upper region of dermis in *Salamandra infraimmaculata* and *Hyla savignyi* (Figure 5A,B). Biotinylated hyaluronic acid binding protein (B-HABP) was used to assess the immunoreactivity of HA. Blocking was performed with 2% bovine serum albumin, and visualization was done with streptavidin-fluorescein isothiocyanate (FITC). For the negative control, hyaluronidase-digested sections were used. The specificity of the staining was controlled by digesting some sections with *Streptomyces* hyaluronidase prior to the incubation with the probe. For the positive control, sections were stained with B-HABP/HA. Based on the former findings mentioned, HA has probably functioned as a water barrier, which provides the transportation of water to the epidermis to maintain skin moisture and decrease the level of mechanical stresses.

## 7. Conclusions

Amphibians through aquatic to terrestrial life are crucial life forms as indicators of environmental well-being. They are one of the most affected taxa from the rapidly changing world and climate. One of the results of these effects is increasing UVB radiation, which brings the biogeochemical processes of aquatic life to a standstill. In relation to this, it has been found that photolyase, a DNA-repairing enzyme after exposure to UVB radiation, increases in exposed embryos. These effects caused the populations to decline rapidly (32.5% threatened (T), 7.4% critically endangered (CR)), faster than birds (12% T, 1.8% CR) and mammals (23% T, 3.8% CR), especially since the 1960s [72].

In conclusion, amphibians, as the bridge of vertebrate evolution, depend on the environmental conditions more than others due to the total water dependence of their embryos and partially terrestrial adult forms. Additionally, a stable surrounding would be required for the complex metamorphosis process, which is regulated by multifarious enzymes and hormones. For this reason, it is essential to monitor and understand the developmental steps of amphibians, which are a micro-reflection of the changes in nature due to their sensitivity, and to conduct new studies.

## Figures and Tables

**Figure 1 jdb-11-00006-f001:**
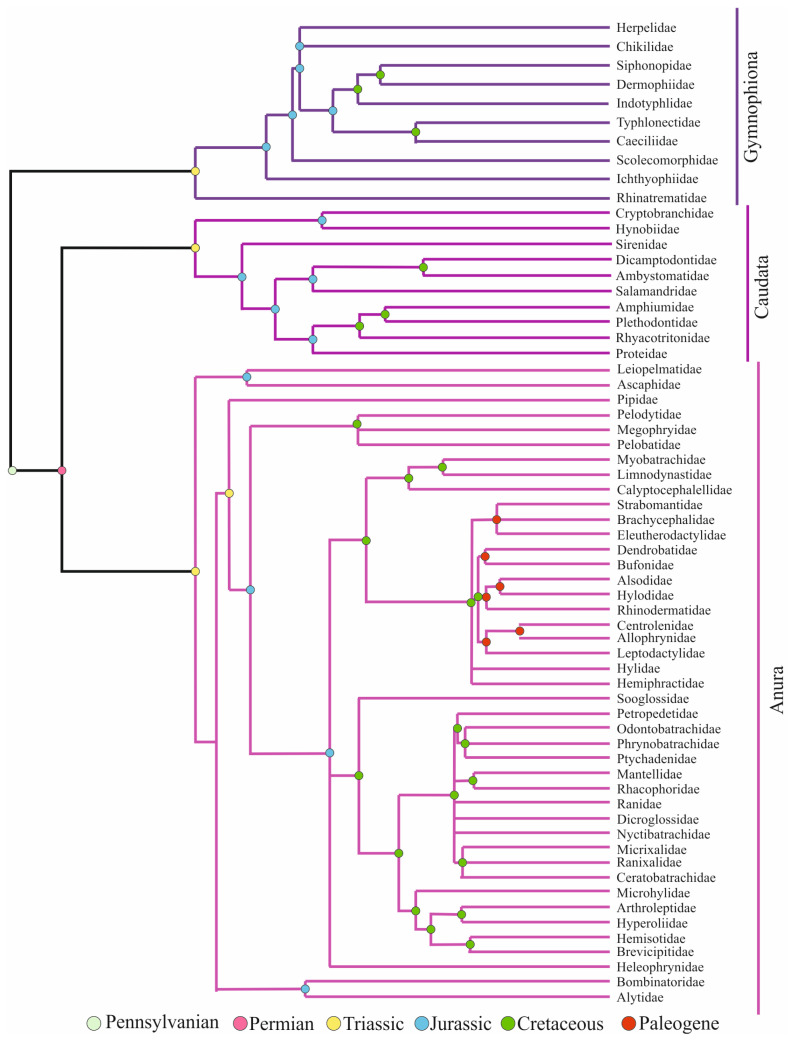
Evolutionary tree of amphibians (adapted from Refs. [3,4,5,6]. 2009, Cannatella, D.C. et al.; 2009, Bossuyt, F. et al.; 2009, Vieties, D.R. et al.; 2009, Gower, D.J. and Wilkinson, M.).

**Figure 2 jdb-11-00006-f002:**
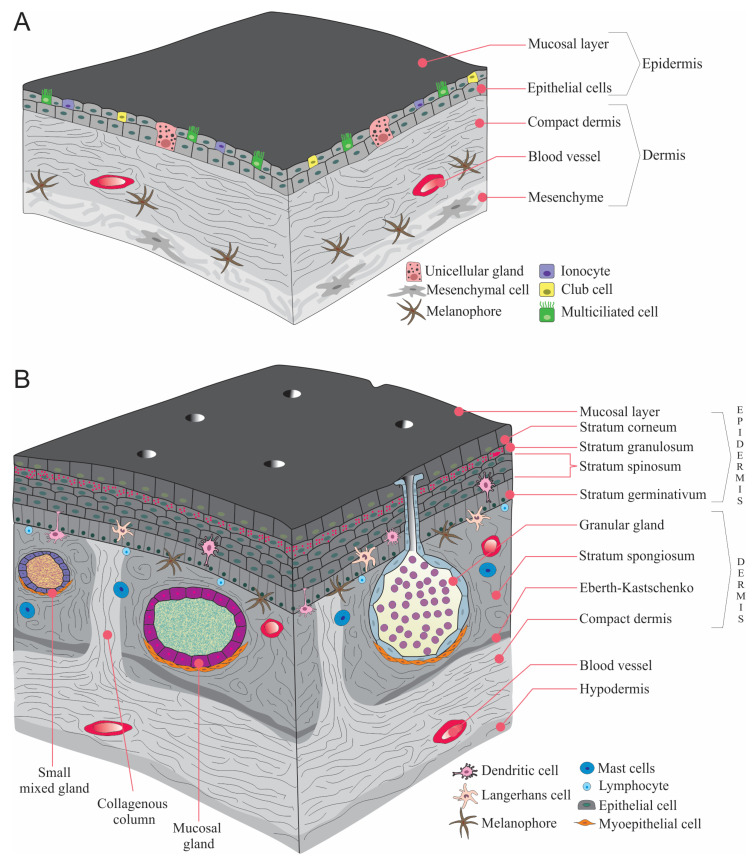
A general illustration of amphibian skin. (**A**). larvae, (**B**). Adult ((**B**) is adapted with permission from Ref. [23]. 2022, Akat et al.).

**Figure 3 jdb-11-00006-f003:**
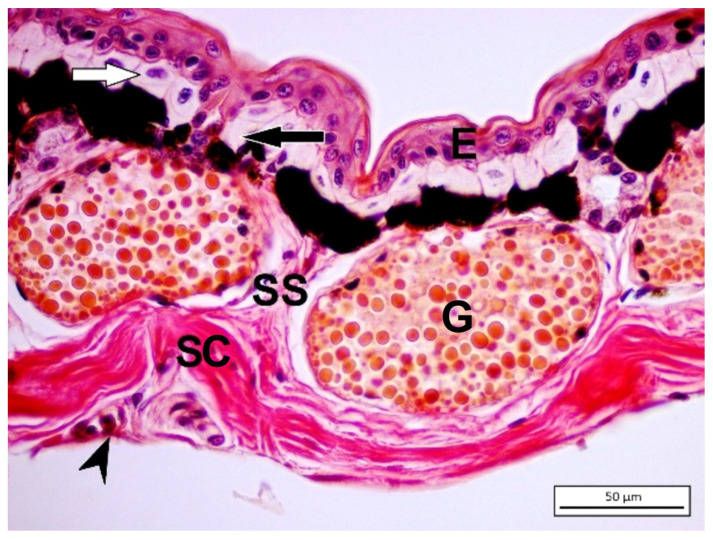
Picro-ponceau staining method on *Hyla savignyi* dorsal skin. (E) Epidermis; (G) granular glands; (SS) stratum spongiosum; (SC) stratum compactum; hypodermis (arrowhead). Rows of chromophores are visible just below the basement membrane. White arrow indicates xanthophore while black arrow indicates iridophores, which are upon melanophores.

**Figure 4 jdb-11-00006-f004:**
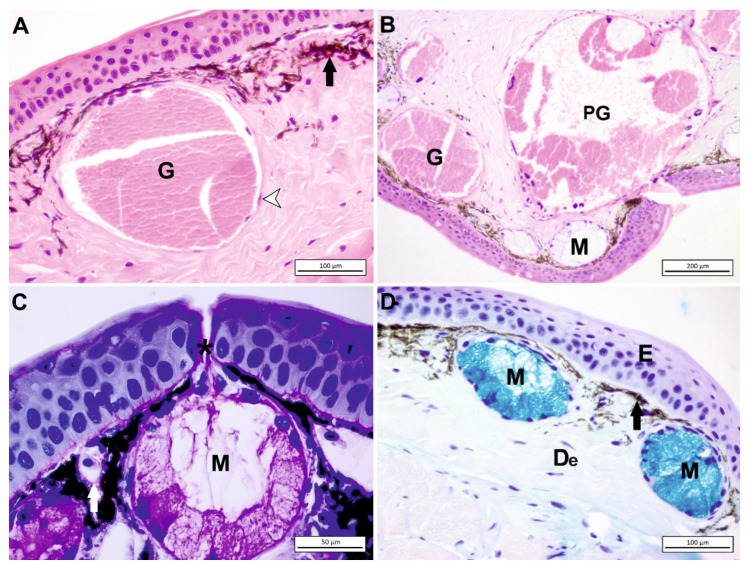
(**A**) The dorsal skin of *Salamandra infraimmaculata*, granular gland (G), melanophore (black arrow). The periphery of dermal gland was surrounded with a monolayer of myoepithelial cells (arrowhead), Gill’s H&E staining. (**B**) Mucous gland (M), granular gland (G), parotoid gland (PG), H&E staining. (**C**) In the dorsal skin of *S. infraimmaculata,* each gland discharges secretion by epidermal duct (*), mucous gland (M), blood vessel (white arrow) PAS staining. (**D**) The ventral skin of *S. infraimmaculata*, epidermis (E), dermis (De), mucous gland (M), melanophore (black arrow) AB staining.

**Figure 5 jdb-11-00006-f005:**
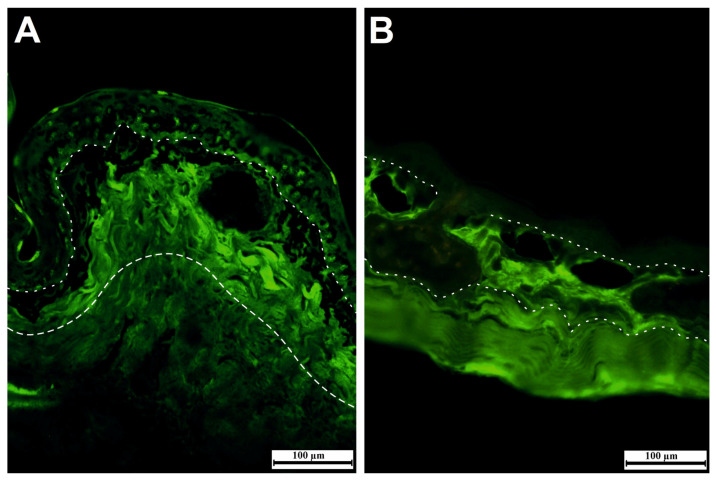
Immunoreactivity of hyaluronic acid (HA) in the dermis of *Salamandra infraimmaculata* (**A**) and *Hyla savignyi* (**B**), mainly in the upper region of the dermis.

**Table 1 jdb-11-00006-t001:** Studies with amphibian species mentioned in the review.

Species	Ordo, Familia	Research Topic
*Rhinella arenarum*	(Anura, Bufonidae)	Development of skin
*Arthroleptis sp.*	(Anura, Arthroleptidae)	Development of skin
*Cardioglossa sp.*	(Anura, Arthroleptidae)	Development of skin
*Hynobius retardatus*	(Urodela, Hynobiidae)	Development of skin
*Hyla savignyi*	(Anura, Hylidae)	Skin histomorphology
*Salamandra infraimmaculata*	(Salamandridae, Salamandridae)	Skin histomorphology
*Hyperolius nasutus*	(Anura, Hyperoliidae)	Chromatophores in the skin
*Hyperolius viridiflavus*	(Anura, Hyperoliidae)	Skin adaptation to wet and dry season conditions
*Dryophytes chrysoscelis* (formerly *Hyla chrysoscelis*)	(Anura, Hylidae)	Expression and immunolocalization of aquaporins
*Chiromantis petersi*	(Anura, Rhacophoridae)	Dermal chromatophores
*Xenopus laevis*	(Anura, Pipidae)	Effects of melatonin during metamorphosis, antimicrobial peptides, aquaporins
*Phyllomedusa bicolor*	(Anura, Hylidae)	Dermal glands
*Bufo sp.*	(Anura, Bufonidae)	Dermal glands
*Lithobates pipiens*	(Anura, Ranidae)	Biochemical and function of epithelial mucus
*Ranoidea splendida* (formerly *Litoria splendida*)	(Anura, Hylidae)	Host-defense peptides of the skin
*Odorrana margaretae*	(Anura, Ranidae)	Antimicrobial peptides of different life stages
*Pelophylax nigromaculatus*	(Anura, Ranidae)	Antimicrobial peptides of different life stages
*Xenopus tropicalis*	(Anura, Pipidae)	The development of the mucosal barrier
*Hyla japonica*	(Anura, Hylidae)	Aquaporins
*Rana japonica*	(Anura, Ranidae)	Aquaporins
*Rana nigromaculata*	(Anura, Ranidae)	Aquaporins
*Rana catesbeiana*	(Anura, Ranidae)	Aquaporins

## Data Availability

The authors confirm that the data supporting the findings of this study are available within the article.

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
