# Peer review of "The Complex Bridge between Aquatic and Terrestrial Life: Skin Changes during Development of Amphibians"

_jdb, 2023, doi:10.3390/jdb11010006_

Round 1

Reviewer 1 Report

This review describes skin structures of tadpole and frog, amphibian chromatophores, skin glands, skin immunology and skin-water relation. Skin changes during development of amphibians, that is a title of the manuscript, is mentioned from anatomical view point. However, this review summarizes amphibian skin biology well.

Minor points

1) Authors should show the position of st. granulosum in Fig. 1.

2) What is SD in Fig.2? It should be SC.

3) Authors identifies iridophore by black arrow in Fig. 2. What is an evidence to demonstrate it?

4) Where is "G" in Fig. 2? 

5) line 166. Figure 2?

6) What is PB in Fig. 3B? Is it PG?

7) Where are "D", "E" and black arrow in Fig. 3D?

8) line 313. Fig. 3 should be Fig. 4.

9) There should be an error in lines 353-354.

10) Authors should show the borderlines between epidermis, stratum spongiosum and stratum compacts by white line or dot line.

Author Response

Response to Reviewers

Reviewer1

Dear Reviewer,

Thank you for your time and critics. We have read and study all your comments carefully and try to edit the article accordingly. Please find our responses to your comments both in the following sentences and within the MS word document. Thank you.

Sincerely,

Esra Akat, Melodi YenmiÅŸ, Berna Çakır

Comments and Suggestions for Authors

This review describes skin structures of tadpole and frog, amphibian chromatophores, skin glands, skin immunology and skin-water relation. Skin changes during development of amphibians, that is a title of the manuscript, is mentioned from anatomical view point. However, this review summarizes amphibian skin biology well.

Minor points

  • Authors should show the position of st. granulosum in Fig. 1.

Authors: We have added the stratum. Thank you.

  • What is SD in Fig.2? It should be SC.

Authors: It has been corrected thank you.

  • Authors identifies iridophore by black arrow in Fig. 2. What is an evidence to demonstrate it?

Authors: Considering the first author's doctoral thesis and literature knowledge on Hyla savignyi skin, iridophores were identified. Three types of chromatophores; melanophores, xanthophores, and iridophores are seen in their dorsal skin and cause the appearance of their characteristic dorsal color. The special localization of three chromatophore types, that is, the xanthophores are located uppermost and just beneath the epidermis, the iridophores are below the xanthophores, and the melanophores are situated undermost.

  • Where is "G" in Fig. 2? 

Authors: It has been added. Thank you.

  • line 166. Figure 2?

Authors: It has been corrected.

  • What is PB in Fig. 3B? Is it PG?

Authors: Yes, corrected. Thank you.

  • Where are "D", "E" and black arrow in Fig. 3D?

Authors: We added the missing values to the figure. Thank you.

  • line 313. Fig. 3 should be Fig. 4.

Authors: Corrected. Thank you.

  • There should be an error in lines 353-354.

Authors: Yes, it has been rewritten. Thank you.

  • Authors should show the borderlines between epidermis, stratum spongiosum and stratum compacts by white line or dot line.

Authors: We added the white lines. Thank you.

Reviewer 2 Report

The manuscript by CÖMDEN et al is focus on skin changes during development of amphibians. This work is really interesting because is an overview of data available on various amphibians. This work needs to be modified and can be accepted after major modification.

Major comments

1- In the summary, author propose to procure information on evolutionary history but this review is not well organized to show that. Authors can do a clear part for this point with clear analyse of evolutionary history (with phylogenetic tree for example ect…).

2- The review is really dense with few illustrations and sometime is hard to follow for the reader. Authors must help the reader with many tables (showing contribution and specificity for each amphibian’s species) and schematics representations.

3- For figure 2. Legend of the figure seams to not correspond to the figure. In legend they have G but not G is present in the figure and they have SC in the legend but SD in figure.

4- For figure 4, is not possible to conclude anything. Experiment with HA labelling need to be developed. It’s really important to show the specificity of immunofluorescence to be sure is not an artefactual labelling. It would be really appreciated if authors show picture with white visible light in addition to fluorescent picture and if they add arrows to help the reader.

Minor points

1- For fig 3: authors can homogenize the legend (A is in bold character but not for B and C). Somethings are written in legend but not show in figure (PG for exemple).

2- This review doesn’t contain material and method. Authors can add one or at least they can detail staining (HE= Hematoxilin/Eosin labelling, AB for…), antibody concentration for HA staining ect…

Author Response

Response to Reviewers

Reviewer 2

Dear Reviewer,

Thank you for your time and critics. We have read and study all your comments carefully and try to edit the article accordingly. Please find our responses to your comments both in the following sentences and within the MS word document. Thank you.

Sincerely,

Esra Akat, Melodi YenmiÅŸ, Berna Çakır

Comments and Suggestions for Authors

The manuscript by CÖMDEN et al is focus on skin changes during development of amphibians. This work is really interesting because is an overview of data available on various amphibians. This work needs to be modified and can be accepted after major modification.

 Major comments

 1- In the summary, author propose to procure information on evolutionary history but this review is not well organized to show that. Authors can do a clear part for this point with clear analyse of evolutionary history (with phylogenetic tree for example ect…).

 Authors: Yes, we have added paragraphs to introduction covering the evolution of amphibians as well as a phylogenetic tree. Thank you.

2- The review is really dense with few illustrations and sometime is hard to follow for the reader. Authors must help the reader with many tables (showing contribution and specificity for each amphibian’s species) and schematics representations.

 Authors: We have added a phylogenetic tree as well as a table that shows all the species we summarized and discussed along with their references and the research topics they’ve been used. Thank you.

3- For figure 2. Legend of the figure seams to not correspond to the figure. In legend they have G but not G is present in the figure and they have SC in the legend but SD in figure.

Authors: The figure and the legends have been corrected. Thank you.

 4- For figure 4, is not possible to conclude anything. Experiment with HA labelling need to be developed. It’s really important to show the specificity of immunofluorescence to be sure is not an artefactual labelling. It would be really appreciated if authors show picture with white visible light in addition to fluorescent picture and if they add arrows to help the reader.

Authors: We have added dashed lines to separate the strata. We also added a paragraph explaining the method under "water relations" subtitle. Since Fig2 and Fig 3D are the LM images of Fig 4B and Fig4A, we did not add new LM images to fig4 here.

 Minor points

1- For fig 3: authors can homogenize the legend (A is in bold character but not for B and C). Somethings are written in legend but not show in figure (PG for exemple).

 Authors: We have homogenized the letters by making them all bold. We also added the missing legends to the figure. Thank you.

2- This review doesn’t contain material and method. Authors can add one or at least they can detail staining (HE= Hematoxilin/Eosin labelling, AB for…), antibody concentration for HA staining ect…

Authors: We added the methods we used related to HA immunoreactivity under "water relations" subtitle. The type of the HE staining has been added to the figure 3 caption. Thank you.

Round 2

Reviewer 2 Report

The authors answered my questions. I consider that the review can be published.